# Complement Inhibitors in Generalized Myasthenia Gravis: Comparison of Administration Schedules, Efficacy, and Safety

**DOI:** 10.3390/jcm14228205

**Published:** 2025-11-19

**Authors:** Giuseppe Di Martino, Nicasio Rini, Alessia Bonaventura, Mauro Trovato, Simona Maccora, Salvatore Maria Lima, Concetta La Seta, Filippo Brighina, Vincenzo Di Stefano

**Affiliations:** 1Department of Biomedicine, Neuroscience and Advanced Diagnostics (BiND), University of Palermo, 90129 Palermo, Italy; 2Unità Operativa Complessa Farmacia, Azienda Universitaria Ospedaliera Policlinico P. Giaccone Hospital, 90129 Palermo, Italy

**Keywords:** complement inhibitor, Myasthenia Gravis, AChR, Eculizumab, Ravulizumab, Zilucoplan

## Abstract

**Background:** Eculizumab, Ravulizumab, and Zilucoplan are inhibitors of terminal complement protein C5 (C5IT) approved for the treatment of generalized Myasthenia Gravis (gMG). The aim of this study is to compare the administration schedules, efficacy, and safety of these new biological therapies in a real-life setting. **Methods:** We enrolled 31 patients with gMG who received C5IT (Eculizumab: 7 patients; Ravulizumab: 11 patients; Zilucoplan: 13 patients). We gathered demographic, clinical data by the difference between scores at baseline (T0) and after follow-up for the MG-ADL, QMG, and MGC scales. **Results:** All C5IT demonstrated similar clinical efficacy, resulting in a statistically significant reduction in clinical scales scores for the MG-ADL (F = 14.7; *p* < 0.001), QMG (F = 14.78; *p* < 0.001), and MGC (F = 9.466; *p* < 0.001), with no significant differences among drugs (*p* > 0.05). No significant differences were highlighted in terms of MSE (*p* > 0.05). There was a decrease in the mean dose of steroid taken by patients in all three treatment groups (Eculizumab: −37%; Ravulizumab: −62%; Zilucoplan: −37%, at W34 compared to baseline). No myasthenic crises requiring hospitalization occurred during follow-up. Most of the reported adverse events were mild to moderate; the more severe events included one case of Stevens–Johnson syndrome (Ravulizumab) and episodes of pneumonia (Eculizumab, Ravulizumab). **Conclusions:** The comparison of C5IT did not bring out significant differences in terms of clinical efficacy and safety, representing a valid therapeutic option when traditional therapies fail to control disease symptoms.

## 1. Introduction

Myasthenia Gravis is an autoimmune disease that affects the post-synaptic membrane at the neuromuscular junction [1]. MG is the most common among neuromuscular junction diseases, caused by the presence of antibodies directed against components of the muscle post-synaptic membrane. The clinical feature is represented by muscle weakness, which typically worsens with repeated muscle exercise. MG is a heterogeneous disease, and the patients can be divided into several subgroups, based on serologic status, clinical phenotype, age at onset, and association with thymic pathology [2]. Analysis of several epidemiological studies has shown that AChR-MG has significantly higher prevalence rates than other serotypes, with incidence rates that appear to be similar globally. Approximately 85% of patients with generalized MG and 50–60% of patients with ocular MG are classified as anti-AChR positive [3,4]. Several studies have suggested that the onset of AChR-MG is correlated to three possible molecular mechanisms: (1) complement activation; (2) antigenic modulation; (3) blockade of AChR function [5]. The complement cascade can be triggered by classical, lectin, or alternative pathways, leading to the formation of proteolytic enzyme complexes, which drive complement activation and amplification. Activation of the classical complement pathway plays a crucial role in AChR-MG. This begins when immunoglobulin molecules belonging to IgG1-3 subclasses and IgM bind C1q. This leads to the activation of C1r and C1s, which cleave C4 and C2, generating C3 convertase. The cleavage of C3 leads to the formation of C5 convertase, responsible for the proteolytic cleavage of C5 into C5a and C5b. This latter fraction drives the assembly of the membrane attack complex (MAC, C5b-9), which causes cell damage and lysis [6]. In this scenario, recently approved molecules, such as complement inhibitors (C5IT), Eculizumab, Ravulizumab, and Zilucoplan, are being used in the pharmacological treatment of gMG. These drugs bind with high affinity and specificity to the complement protein C5 and prevent association between C5 convertase and C5, thus inhibiting the subsequent cleavage of C5 into C5a and C5b and ultimately the formation of the MAC. Additionally, by binding to the C5b, Zilucoplan sterically hinders binding of C5b to C6, which prevents the subsequent assembly and activity of the membrane attack complex, thereby reducing complement-mediated damage at the neuromuscular junction [7,8,9]. The study conducted includes a prospective, longitudinal, single-center analysis in a real-life setting aimed at comparing the administration schedules, efficacy, and safety of C5IT used in the treatment of gMG.

## 2. Patients and Methods

### 2.1. Study Design and Inclusion Criteria

The study was approved by the competent Ethical Committee “Palermo I”, protocol code 6/2022, approved on 14 June 2022, and it was conducted in conformity with the Declaration of Helsinki principles. Written informed consent was obtained from all patients for the use of anonymized clinical data for research purposes. We conducted an observational and retrospective study to evaluate the safety and efficacy of C5IT in patients with gMG in a real-life clinical setting.

Patients were considered eligible if they met all the following criteria:Initiation of treatment with Eculizumab, Ravulizumab, or Zilucoplan, as part of clinical practice.Age ≥ 18 years at the time of informed consent and treatment initiation.MG diagnosis in accordance with national guidelines and based on clinical, serological (positive anti-AChR antibody test), and neurophysiological (positive repetitive nerve stimulation test and/or single-fiber electromyography) criteria.Myasthenia Gravis Foundation of America (MGFA) classification IIa (only for Zilucoplan, according to prescription requirement in Italy), IIb, IIIa, IIIb, IVa, and IVb.Post-intervention status unchanged or worsened after treatment with corticosteroids and at least one other non-steroid immunosuppressant (NSIST), administered at adequate dosages and for an adequate duration, but with persistent symptoms or side effects that impaired functionality, as assessed by both the patient and the treating physician.

### 2.2. Data Collection

All patients were recruited at the Rare Neuromuscular Disease Clinic of the A.O.U.P. “P. Giaccone” in Palermo, between October 2022 and July 2025. We collected data using local databases that served as source data and further processed for data cleaning and analysis. The study protocol required patient enrolment at T0, corresponding to the day of initiation of therapy with Eculizumab, Ravulizumab, or Zilucoplan. The choice of one drug over another was determined by the availability of alternative therapies, the presence of inclusion criteria specified in the drug’s official data sheet and approved by the regulatory authority, and the patient’s preference regarding the administration schedule. At T0, participants were administered a standardized evaluation including MG-ADL, QMG, and MGC scales [10,11,12]. Scores were recorded on specific forms and updated at each follow-up. Follow-up assessments, according to the previously reported scales, were performed concurrently with the administration of biologic treatment in a day-hospital setting or—where applicable—according to a home-based treatment schedule defined during the maintenance phase. The following variables were included in the data collection: current age, sex, and weight; comorbidities; age at onset of MG and age at treatment initiation; disease duration at baseline; MGFA clinical classification at disease onset and at baseline; history of thymectomy and thymic status; previous treatments, including prednisone, NSIST, and chronic IVIg or PLEX; history of prior myasthenic exacerbations and/or myasthenic crisis (MC), hospitalizations, and any use of rescue therapies. Previous and baseline dosages of prednisone and NSIST were recorded. During C5IT therapy, any changes in the dosages of ongoing baseline medications, adverse events, treatment discontinuation, and the use of rescue therapies were also recorded. An age of 50 years of age at disease onset was used as the cut-off between early-onset MG (EOMG) and late-onset MG (LOMG) [2]. Comorbidities included hypertension, diabetes mellitus, thyroid disorders, cardiovascular disease, chronic respiratory disease, autoimmune or rheumatologic disorders, gastrointestinal diseases, osteoporosis, and other neurological conditions according to the previous literature [13].

### 2.3. Treatment Schedules and Vaccinations

According to the national healthcare system’s reimbursement criteria, eligibility for C5IT included AChR antibody positivity, MGFA ≥ III (for Eculizumab) and ≥II (for Ravulizumab and Zilucoplan), total MG-ADL score ≥ 6, and at least one of the following despite standard treatment: ≥1 myasthenic crisis or major exacerbation event per year requiring IVIg or plasma exchange; need for periodic IVIg or PLEX; intolerable side effects/comorbidities that limit or contraindicate the use of immunosuppressants. Thymectomy in the last 12 months was an exclusion criterion for study inclusion.

All treatments were administered according to the dosages indicated in the product information sheets [7,8,9]. For Eculizumab, the dosage regimen for the treatment of gMG consists of a 4-week induction phase followed by a maintenance phase. The induction phase consisted of four weekly and consecutive IV doses of 900 mg. The maintenance phase consisted of 1200 mg intravenous infusion at the fifth week, followed by 1200 mg intravenous infusion every 14 days ± 2 days.

Ravulizumab was administered intravenously according to the patient’s weight, with an initial loading dose of 2400 mg, 2700 mg, or 3000 mg. This was followed by a second infusion two weeks later, consisting of a maintenance dose of 3000 mg, 3300 mg, or 3600 mg. Subsequent infusions were administered every 8 weeks ± 7 days.

Zilucoplan was administered by daily subcutaneous self-injection with a dose of 0.3 mg/kg (16.6 mg, 23 mg, or 32.4 mg depending on body weight).

All patients were vaccinated against Neisseria meningitidis serogroups A, C, W135, Y, and B, with vaccination occurring at least 14 days before the first dose of treatment. Alternatively, patients received appropriate prophylactic antibiotics before treatment initiation and continued until the completion of the vaccination regimen.

### 2.4. Clinical Scales and Outcome Measures

#### 2.4.1. Myasthenia Gravis Activity of Daily Living (MG-ADL)

The MG-ADL is a self-report-based assessment tool developed to provide a rapid measurement of the frequency and severity of MG symptoms. Through 8 items covering the previous 7 days, it assesses aspects of speech, chewing, swallowing, breathing, muscle strength of the upper and lower limb, as well as diplopia and ptosis. For each item a score is assigned from 0 to 3, for a total score ranging from 0 (absence or minimal presence of symptoms) to 24 (severe symptoms) [10].

#### 2.4.2. Quantitative Myasthenia Gravis Score (QMG)

The QMG is a score that quantifies the severity of MG in relation to the degree of impairment of body functions and structures as indicated in the Classification of Functioning, Disability and Health (WHO 2001). The QMG includes 13 items designed to assess ocular, bulbar, and limb function. For each item a score ranging from 0 to 3 is assigned, for a maximum total of 39, which indicates greater impairment. This assessment scale, based on clinical examination, uses quantitative tests to study specific muscle groups and requires minimal equipment, including a spirometer with mouthpieces, nose clips, chronometer, water and glasses, and dynamometer [11].

#### 2.4.3. Myasthenia Gravis Composite Scale (MGC)

The MGC is a reliable and valid tool for measuring the clinical status of MG patients in clinical practice and studies. The MGC is easy to administer, taking just a few minutes to complete, and no equipment is required. It consists of 10 items, derived from other scales, which assign scores to ptosis, diplopia, eye closure, chewing, swallowing, breathing, neck flexion, shoulder abduction, and leg flexion [12].

### 2.5. Outcomes

Data were retrospectively collected at baseline and at each time point, up to the most recent follow-up. Minimal symptom expression (MSE) as defined by MG-ADL ≤ 1 was assessed for each patient at follow-up end. Changes in MG-ADL and QMG scores over time were the primary outcome measures of the study to evaluate the efficacy of C5IT therapy. Patients were classified as responders if they demonstrated a reduction of at least 3 points in the MG-ADL score (MG-ADL responders) and a reduction of at least 5 points in the QMG score (QMG responders) at each time point of the follow-up. The responder rate of 3 points for MG-ADL and 5 points for QMG were analyzed to ensure comparability among trials, where these stricter cutoffs were used to define clinically meaningful improvement [7,8,9].

Secondary outcome measures included tapering of prednisone during follow-up, as well as the comparison of the need for IVIg or PLEX cycles in the year before and during C5IT treatment. Safety was evaluated by monitoring and documenting treatment-emergent adverse events throughout the study period. Exacerbation, as well as eventual use of rescue therapies, and myasthenic crisis have been registered from baseline and for each follow-up.

### 2.6. Statistical Analysis

We provide a descriptive analysis of demographic and baseline clinical characteristics. Categorical variables were summarized as frequency and percentage, whereas continuous variables were reported as mean ± standard deviation (SD) or median and interquartile range (IQR), as appropriate. Unless otherwise stated, 95% confidence intervals (CIs) of the mean were calculated to allow comparison with previously published data. The distribution of the data was assessed using the Shapiro–Wilk test, which is considered more reliable for the small sample size (*n* < 50). The analysis confirmed the normality of the distribution of the rating scale scores. Differences between treatment groups over time for the MG-ADL, QMG, and MGC scores for the prednisone dose were analyzed using the Linear Mixed Model (LMM), given the longitudinal design of the study and the absence of some data due to an insufficiently extended follow-up for all patients. The effect of time was used to estimate the overall reduction in scores across the different scales during treatment, while the combined time*treatment effect was used to estimate the impact of each treatment. F-tests were reported for each comparison. The minimal clinically important difference (MCID) [14] on the MG-ADL scale was set to 2 points, while on the QMG it was set to 3 points. Comparisons of responder rates and MSE between the treatment groups were performed using the chi-square test and Fisher’s exact test, as appropriate. Odd Ratio (OR) was calculated for each time point. All statistical tests were two-tailed, with significance set at *p* < 0.05. Analysis was performed using JAMOVI version 2.7.6.0 (Sydney, Australia). Graphs and tables were created using SPSS version 31.0.0.0 (IBM Corp, Armonk, NY, USA) and Microsoft Excel (Redmond, WA, USA). We compared treatment effects up to 42 weeks from baseline since after this time point the number of followed patients decreased, limiting the generalizability of the results.

## 3. Results

### 3.1. Characteristics of Enrolled Patients

Table 1 reports clinical features of included patients in this study.

We included overall 31 patients treated with Eculizumab (*n* = 7), Ravulizumab (*n* = 11), and Zilucoplan (*n* = 13), enrolled between October 2022 and July 2025. The first patient treated with Eculizumab started therapy in October 2022, the first patient with Ravulizumab in April 2024, and the first patient with Zilucoplan in November 2023. Of the 31 patients, 68% (*n* = 21) were female and 32% (*n* = 10) were male. The mean age at MG diagnosis was 51.8 ± 18.1 years (Eculizumab = 63.9 ± 13 years; Ravulizumab = 52.4 ± 16.3 years; Zilucoplan = 42.9 ± 19.4 years). Thirteen patients were classified as EOMG (Eculizumab = 1; Ravulizumab = 4; Zilucoplan = 8), while the remaining eighteen were classified as LOMG (Eculizumab = 6; Ravulizumab = 7; Zilucoplan = 5). The mean disease duration was 7.2 ± 7.6 years (Eculizumab = 2.4 ± 1.3 years; Ravulizumab = 7.5 ± 9.4 years; Zilucoplan = 9.46 ± 7.1 years). The mean age at baseline was 59 ± 15.3 years (Eculizumab = 66.3 ± 12.9 years; Ravulizumab = 59.8 ± 14.1 years; Zilucoplan = 54.4 ± 16.8 years). At enrollment, 84% of patients presented with at least one concomitant comorbidity. The prevalence of multimorbidity, at T0, was attested at 86% in the Eculizumab group, 64% in the Ravulizumab group, and 100% in the Zilucoplan group. Twelve patients (39%) had undergone thymectomy, and among them seven (58%) had histopathological diagnosis of thymoma.

All patients included in the study tested positive for anti-AChR antibodies, and none tested positive for anti-MuSK antibodies. We stratified patients at T0 according to the MGFA scale: Eculizumab (*n* = 7): 2 MGFA IIIA, 4 MGFA IIIB, 1 MGFA IVB; Ravulizumab (*n* = 11): 3 MGFA IIB, 1 MGFA IIIA, 6 MGFA IIIB, 1 MGFA IVB; Zilucoplan (*n* = 13): 2 MGFA IIA, 5 MGFA IIB, 2 MGFA IIIA, 4 MGFA IIIB.

### 3.2. MG-ADL

Treatment with all three innovative drugs significantly reduced the MG-ADL score (F = 14.7; *p* < 0.001). The difference was already significant after two weeks (F = 67.9; *p* < 0.001) and remained significant after 4 weeks (F = 70.7; *p* < 0.001), 10 weeks (F = 60.9; *p* < 0.001), 18 weeks (F = 67.6; *p* < 0.001), 26 weeks (F = 60.6; *p* < 0.001), 34 weeks (F = 46.6; *p* < 0.001), and 42 weeks of treatment (F = 41.1; *p* < 0.001). Mean MG-ADL values reduced for Eculizumab from 12.4 ± 1.9 (T0) to 5.5 ± 4.2 (W42); for Ravulizumab from 10.4 ± 3.8 (T0) to 5.8 ± 3.5 (W42); and for Zilucoplan from 9.7 ± 2.8 (T0) to 2.4 ± 1.7 (W42). No significant differences were found when comparing patients treated with Eculizumab, Ravulizumab, and Zilucoplan depending on the entity of the reduction in the MG-ADL score over the observation period (F = 1.173; *p* = 0.324; Figure 1).

When comparing treatment responders, a significant difference in the number of patients who achieved MCID (Δ2 on MG-ADL scale) was observed at W4 between Zilucoplan and Ravulizumab (*p* = 0.024), although the calculated OR showed a very wide confidence interval (OR: 18.7; IC 95%: 0.87–401.8), indicating high uncertainty of the estimate. At the same follow-up, no statistically significant differences emerged between Zilucoplan–Eculizumab (*p* = 0.350) and Eculizumab–Ravulizumab (*p* = 0.338). Furthermore, no significant differences were observed at W10 (*p* = 0.467), W18 (*p* = 1.000), W26 (*p* = 0.555), W34 (*p* = 0.571), and W42 (*p* = 0.158). Setting the cut-off definition of MG-ADL responders at a reduction of at least 3 points from baseline (responder threshold), no significant differences were found between the three treatments over the studied period.

### 3.3. QMG

The treatments significantly reduced the QMG score over the observation period (F = 14.78; *p* < 0.001).

The difference was already significant after two weeks (F = 55.3; *p* < 0.001) and remained significant after 10 weeks (F = 57.1; *p* < 0.001), 18 weeks (F = 60.3; *p* < 0.001), 26 weeks (F = 49.9; *p* < 0.001), 34 weeks (F = 50.3; *p* < 0.001), and 42 weeks of treatment (F = 32.4; *p* < 0.001). Mean QMG values reduced for Eculizumab from 18.3 ± 5.9 (T0) to 11 ± 8.8 (W42); for Ravulizumab from 14.2 ± 5.3 (T0) to 8.3 ± 2.7 (W42); and for Zilucoplan from 15.9 ± 5.9 (T0) to 7.3 ± 3.2 (W42). No significant differences were found when comparing patients treated with Eculizumab, Ravulizumab, and Zilucoplan, regarding QMG score reduction during the studied period (F = 1.005; *p* = 0.379, Figure 2).

When considering a change of ≥3 points in the total QMG score from baseline (MCID), no statistically significant differences were found between Eculizumab, Ravulizumab, and Zilucoplan in responder rate at W02 (*p* = 0.328), W10 (*p* = 0.878), W18 (*p* = 1.000), W26 (*p* = 0.853), and W34 (*p* = 0.649). A significant difference emerged at W42 between Ravulizumab and Eculizumab (*p* = 0.033); however, the remarkably wide confidence interval of the OR indicates a high degree of uncertainty of the estimate due to the small number of patients at the specific follow-up (OR 30.333; IC 95%: 0.9588–959.66). No statistically significant differences were observed when considering a change of ≥ 5 points in the QMG (responder threshold) score from baseline (*p* = 1.000).

### 3.4. MGC

The treatments significantly reduced the MGC score in the observed period (F = 9.466; *p* < 0.001). The reduction in the MGC was already significant at two weeks (F = 25.4; *p* < 0.001) and remained significant at 10 weeks (F = 45.6; *p* < 0.001), 18 weeks (F = 41.7; *p* < 0.001), 26 weeks (F = 37.6; *p* < 0.001), 34 weeks (F = 32.3; *p* < 0.001), and 42 weeks of treatment (F = 25; *p* < 0.001). Mean MGC values reduced for Eculizumab from 21.6 ± 7.2 (T0) to 15.7 ± 8.1 (W42); for Ravulizumab from 17 ± 7.7 (T0) to 8.3 ± 2.7 (W42); and for Zilucoplan from 16.8 ± 7.8 (T0) to 3.3 ± 3.2 (W42). No statistically significant differences were found between groups of patients treated with different therapies (F = 1.644; *p* = 0.213, Figure 3).

Comparing MGC scores among therapies, a progressive decrease in the score is seen up to week 26, associated with a progressive clinical improvement, which continues at week 34 for Ravulizumab and up to week 42 for Zilucoplan. The group treated with Eculizumab saw an increase in the mean MGC from week 26 to week 34, a finding likely related to the small number of patients for that specific follow-up. For Ravulizumab and Zilucoplan, the largest variation occurs between week 2 and week 10; the former reaches its nadir at week 34 and the latter at week 42. For the group treated with Eculizumab, the largest variation in the mean MGC score occurs in the first two weeks, reaching its nadir at week 26. It is noteworthy that, since the T0, the MGC scores for Eculizumab have consistently been higher than those found for the other two groups; this may be related to a greater clinical impairment in these patients compared to those in the Ravulizumab and Zilucoplan patients. Figure 4 shows the responder rates of the MG-ADL and QMG, respectively.

### 3.5. Minimal Symptom Expression

In agreement with data in the literature, a cut-off of MG-ADL ≤ 1 was identified to define minimal symptom expression (MSE).

MSE in Eculizumab: W2: 0; W18: 1 (14%); W26: 1 (14%); W34: 1 (14%); W42: 1 (14%).MSE in Ravulizumab: W2: 1 (9%); W18: 1 (9%); W26: 3 (27%); W34: 2 (18%); W42: 1 (9%).MSE in Zilucoplan: W2: 0; W18: 2 (15%); W26: 2 (15%); W34: 2 (15%); W42: 4 (31%).

No significant differences were observed at W2 (*p* = 0.581), W18 (*p* = 1.000), W26 (*p* = 0.599), W34 (*p* = 1.000), or W42 (*p* = 0.695) in terms of the number of patients in MSE in the groups treated with Eculizumab, Ravulizumab, and Zilucoplan.

### 3.6. Steroid-Sparing Effect

Figure 5 shows the steroid-sparing effect in patients treated with C5IT.

It was also possible to evaluate the steroid-sparing effect of the three C5IT, considering the baseline prednisone dosage and any changes during patients’ follow-ups. All patients included in the Eculizumab, Ravulizumab, and Zilucoplan groups underwent corticosteroid therapy before starting treatment with the respective biologics. Of these, five patients in the Eculizumab group were taking prednisone at baseline, with a mean daily prednisone dosage of 19.3 ± 26.4 mg; eight patients who started treatment with Ravulizumab were taking prednisone at baseline, with a mean daily prednisone dosage of 13 ± 13 mg; twelve patients in the Zilucoplan group were receiving daily prednisone therapy at baseline with a mean dose of 12.65 ± 7.57 mg. The graph in Figure 4 shows, for Eculizumab, Ravulizumab, and Zilucoplan, a significant reduction over time in the mean dose of prednisone taken by patients, of −37%, −62%, and −37%, respectively, at week 34 compared to baseline (we preferred, in this case, to stop our observation at week 34, as in subsequent follow-ups the number of patients receiving prednisone was too small for a comparison), with no statistically significant differences between the three groups. The mean daily dose of prednisone at week 34 was 18.13 ± 22.10 mg for patients receiving Eculizumab; 5.00 ± 5.70 mg for patients treated with Ravulizumab; and 7.92 ± 7.28 mg in patients in the Zilucoplan group. Furthermore, regarding Eculizumab, of the five patients who were taking prednisone at baseline, one of them discontinued prednisone therapy; one patient treated with Ravulizumab discontinued prednisone therapy during follow-up; two patients in the Zilucoplan group discontinued prednisone.

### 3.7. Thymectomy

Twelve patients (39%) had previously undergone thymectomy and seven patients (23%) had a histological diagnosis of thymoma. Thymectomy was performed prior to the start of C5IT therapy in all patients. The interval between thymectomy and C5IT initiation ranged from 2.1 to 22.6 years, with a median of 5.4 years (IQR: 3.4–9.0 years). In the thymoma-associated MG subgroup, the interval between thymectomy and C5IT initiation ranged from 2.1 to 16.8 years, with a median of 4.6 years (IQR: 2.9–9.0 years).

### 3.8. Safety and Adverse Events

One of the objectives of our study was also to analyze and compare the safety profiles of Eculizumab, Ravulizumab, and Zilucoplan. Table 2 reports the adverse events in patients included in this study.

Of the seven patients treated with Eculizumab, three (43%) experienced adverse events potentially related to drug administration, including headache (14%), diarrhea (14%), and pneumonia (14%). In the Ravulizumab treatment group, five patients (45%) developed adverse events, mostly mild in severity: nausea (27%), fever (18%), and general malaise (9%). In addition, one episode of pneumonia and one case of Stevens–Johnson syndrome were recorded in a patient treated with Ravulizumab. Finally, in the Zilucoplan group, eight patients (62%) experienced adverse events, mostly mild, associated with drug administration: injection-site reaction (62%), headache (31%), and nausea (15%).

No myasthenic crisis requiring immediate hospitalization occurred during follow-up. One patient treated with Eculizumab and two patients treated with Zilucoplan experienced disease exacerbation during follow-up, of which one required rescue therapy (IVIg/PLEX). Three patients receiving Eculizumab did not reach week 42 of follow-up. One died due to complications unrelated to the study treatment; the other one discontinued therapy following hospitalization for peritonitis complicated by intestinal perforation, while another discontinued due to worsening symptoms and poor control of the disease despite treatment. In the Ravulizumab group, two patients discontinued therapy due to a lack of benefit and poor symptom control. Three patients treated with Zilucoplan discontinued therapy: one due to recurrent infections requiring hospitalization, another because of the onset of an unspecified rheumatologic condition, and another following an episode of morphea that required prolonged hospitalization.

## 4. Discussion

This study compared the clinical efficacy and safety of three C5IT—Eculizumab, Ravulizumab, and Zilucoplan—in patients with gMG in a real-world clinical setting. Although all three agents share a common mechanism of complement inhibition, they have three different administration schedules which may influence patient preference, adherence, and treatment burden [15]. Subcutaneous administration of Zilucoplan is certainly an advantage as it represents a less demanding alternative for patients, who can self-administer the therapy at home [16]. At the same time, the longer infusion interval of Ravulizumab (every 8 weeks) represents practical benefits compared to the biweekly infusions required with Eculizumab [17,18]. These characteristics may improve long-term adherence, reduce hospital resource use, and enhance patient satisfaction. Through standardized, periodic evaluations, we were able to compare treatment outcomes across the three agents. No significant differences emerged among the drugs in terms of reduction in the MG-ADL, QMG, and MGC scores; indeed, all three treatments led to a clear clinical improvement with a comparable time profile.

### 4.1. Comparison with Previous Clinical Trial

The clinical improvements observed in our cohort were rapid and sustained across the MG-ADL, QMG, and MGC scores. Reductions in the MG-ADL and QMG were already significant at early time points and remained stable during follow-up. Our study showed that the absolute reduction in the MG-ADL score was greater than that previously reported in the REGAIN study for Eculizumab (−5.8 vs. −4.1) [7]. This reduction was comparable to that found in the CHAMPION-MG study (−4.00 vs. −3.8) for Ravulizumab and for Zilucoplan compared to that found in the RAISE study (−4.38 vs. −4.39) [8,9]. Similarly, all three agents also led to a significant reduction in the QMG score after the first week of treatment. From this point of view, the improvement in patients treated with Eculizumab and Ravulizumab stabilized after the tenth week; for the group of patients treated with Zilucoplan, the reduction in the mean of the QMG score also persisted during the subsequent follow-ups. The reduction in the QMG score was greater than previously reported in the REGAIN study for Eculizumab (−6.71 vs. −4.6) [7]. This reduction was also greater for Ravulizumab than in the CHAMPION-MG study (−4.63 vs. −2.8) and slightly lower for Zilucoplan than in the RAISE study (−5.08 vs. −6.19) [8,9]. Consistent with these findings, MGC scores also declined significantly over time in all treatment groups. Overall, our results indicate a higher response rate than those reported in phase III trials for both the MG-ADL and QMG scales. 

Overall, clinical responses in our real-world cohort were comparable to those reported in phase III trials. Responder rate analysis revealed that ≥50% of patients achieved the MCID for the MG-ADL (≥2-point improvement) by week 4 and continued to increase over time. When applying a more stringent criterion (≥3-point improvement), responder rates ≥50% were still observed for all three treatments after week 4, in line with the REGAIN, CHAMPION-MG, and RAISE studies [7,8,9]. Analyzing responder rates based on changes in the QMG scores, all study groups showed rates ≥50% from week 10 when considering a ≥3-point improvement from the baseline (MCID). In this case, Ravulizumab and Zilucoplan appeared to achieve a faster response, even from the second week of treatment, compared to Eculizumab. When considering a ≥5-point improvement in total QMG score from baseline (responder threshold in REGAIN, CHAMPION-MG, RAISE), the proportion of responders increased more slowly compared to the previous cut-off for all three treatments, never reaching 100% at any of the time points evaluated in our study.

Minimal symptom expression (MG-ADL ≤ 1) was achieved by at least one patient in each treatment group during follow-up, with a percentage of 37.5% for Ravulizumab at week 26, and 25% for Eculizumab and 44.4% for Zilucoplan at week 42.

### 4.2. Steroid-Sparing Effect

All three C5IT were associated with a progressive steroid-sparing effect. The mean prednisone dosage decreased across treatment groups, most prominently with Ravulizumab (−62%) and comparable for Eculizumab and Zilucoplan (−37%) from baseline to week 34. This finding highlights a clinically significant steroid-sparing effect of C5IT, which is associated with a lower incidence of side effects attributable to chronic steroid therapy [19,20]. Our findings therefore support the use of complement inhibitors not only for symptom control but also as steroid-sparing agents that improve overall tolerability and quality of life.

### 4.3. Safety and Tolerability

The safety profile of the three C5 inhibitors was consistent with previous reports [18,19,20,21,22,23]. Most adverse events were mild or moderate, resolved spontaneously or with symptomatic treatment, and had no long-term impact on patients’ quality of life. More severe events included one case of Stevens–Johnson syndrome (in a patient treated with Ravulizumab, already included in the study by Rossini et al. [19]) and a few episodes of pneumonia requiring hospitalization, all of which were resolved with appropriate medical management. These results confirm the favorable safety and tolerability profile of C5IT in gMG.

### 4.4. Limitations

The limitations encountered in the study are primarily related to the small sample size and the lack of homogeneity within and between treatment groups. Quantitatively, different groups were compared, and the patients included in the study had different MG severity, age, and disease duration, as well as comorbidity. C5IT were administered as add-ons, with patients continuing their baseline treatment (CS and NSIST), thus adding another element of heterogeneity. Not all patients were receiving CS when they were included in the study, and patients receiving steroid therapy did not all receive the same dosage at the baseline and at subsequent follow-ups. Furthermore, patients treated with immunosuppressants (84%) did not all receive the same type of immunosuppressant and at the same dosage. The number of patients enrolled in the study progressively decreased during follow-up. Indeed, not all patients reached week 42 of treatment: some patients discontinued therapy due to a lack of symptom control or other concomitant conditions that required treatment discontinuation. The results should be considered preliminary and must be confirmed through larger multicenter studies with standardized outcome measures.

## 5. Conclusions

This study provides a real-world comparison of the three available C5 inhibitors in generalized Myasthenia Gravis, representing a novel contribution to the current literature and the first real-life data on Zilucoplan. Overall, our findings confirm the rapid and sustained clinical benefit of C5 inhibition, its steroid-sparing, and its favorable safety profile, reinforcing the role of complement-targeted therapy in the management of gMG. Further studies with larger cohorts and longer follow-up periods are needed to better define comparative efficacy, safety, and patient-reported outcomes among the available C5IT for gMG.

## Figures and Tables

**Figure 1 jcm-14-08205-f001:**
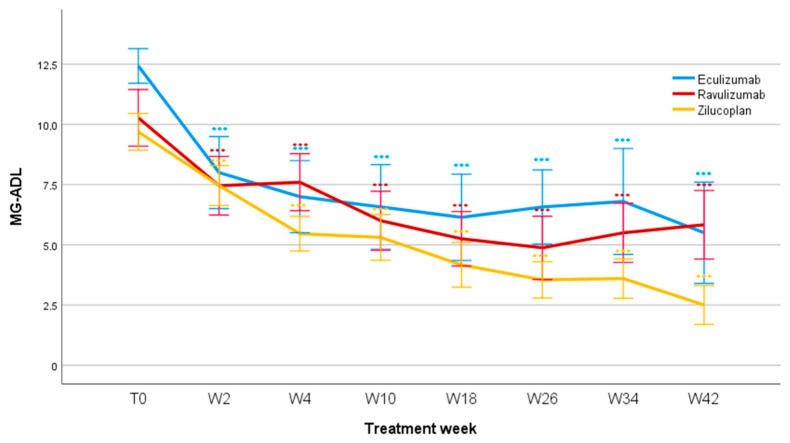
MG-ADL scores during follow-up. MG-ADL, Myasthenia Gravis Activity of Daily Living; T0, baseline; W, week. *** *p* < 0.001.

**Figure 2 jcm-14-08205-f002:**
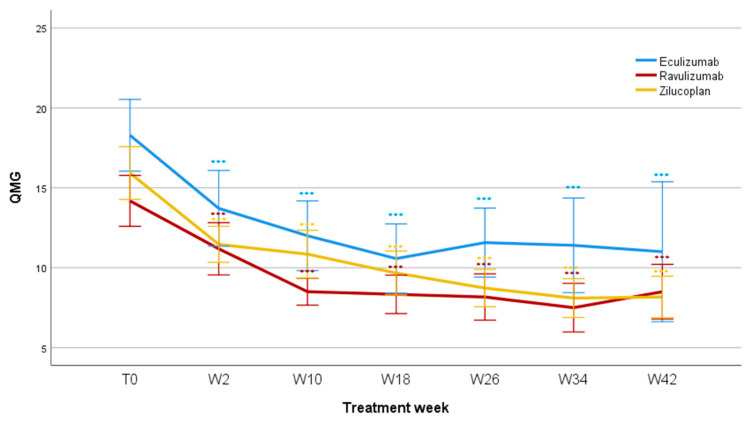
QMG scores during follow-up. QMG, Myasthenia Gravis quantitative scale; T0, baseline; W, week; *** *p* < 0.001.

**Figure 3 jcm-14-08205-f003:**
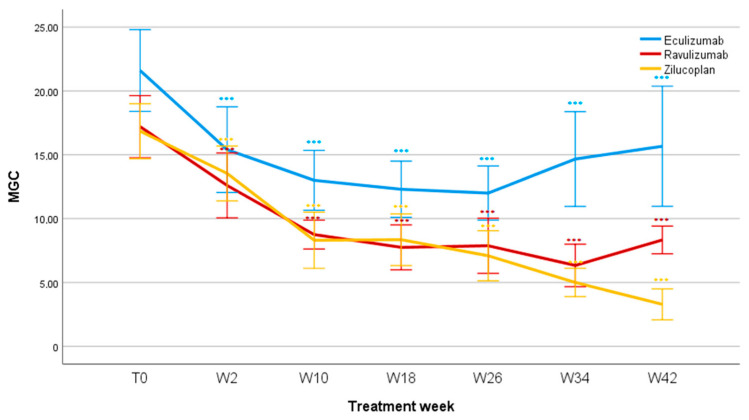
MGC scores during follow-up. MGC, Myasthenia Gravis Composite; T0, baseline; W, week. *** *p* < 0.001.

**Figure 4 jcm-14-08205-f004:**
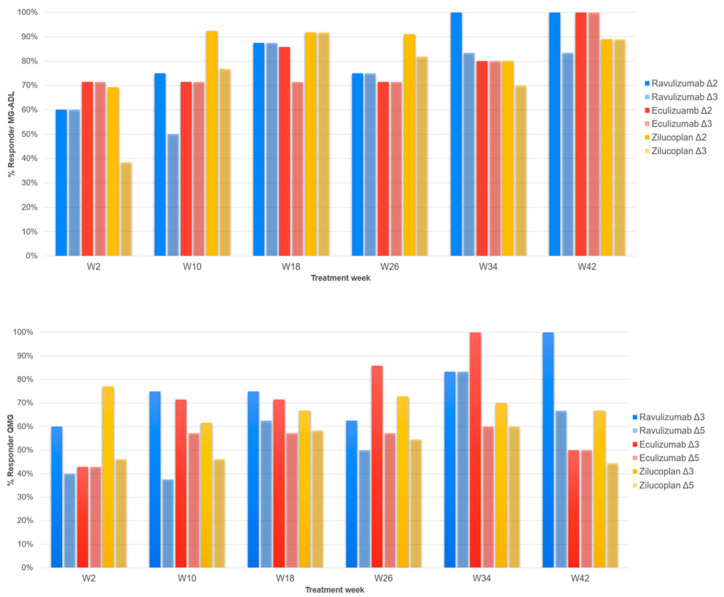
The upper figure shows the responder rate of the MG-ADL scale. Two different color tones indicate the different cut-offs (Δ2: a difference ≥ 2 points from T0; Δ3: a difference ≥ 3 points from T0) for each treatment, as reported in legend. W2, 2 weeks of follow-up; W10, 10 weeks; W18, 18 weeks; W26, weeks; W34, 34 weeks; W42, 42 weeks. At the bottom, the responder rate of the QMG scale is shown. Two different color tones indicate the different cut-offs (Δ3: a difference ≥ 3 points from T0; Δ5: a difference ≥ 5 points from T0) for each treatment, as reported in legend. W2, 2 weeks of follow-up; W10, 10 weeks; W18, 18 weeks; W26, weeks; W34, 34 weeks; W42, 42 weeks.

**Figure 5 jcm-14-08205-f005:**
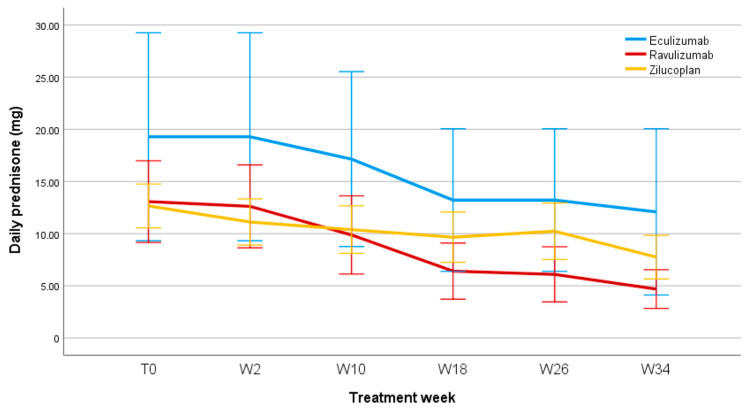
Daily prednisone dose (mg) during follow-up. T0, baseline; W, week.

**Table 1 jcm-14-08205-t001:** General demographics.

	Eculizumab	Ravulizumab	Zilucoplan	Total
Patients (*n*, %)	7 (23%)	11 (35%)	13 (42%)	31 (100%)
Female (*n*, %)	5 (71%)	5 (45%)	11 (85%)	21 (68%)
Male (*n*, %)	2 (29%)	6 (55%)	2 (15%)	10 (32%)
Age (years, mean)	66.3	59.8	54.4	59
Age at Diagnosis (years, mean)	63.9	52.4	44.9	51.8
Disease Duration (years, mean)	2.4	7.4	9.46	7.2
EOMG (*n*, %)	1 (14%)	4 (36%)	8 (62%)	13 (42%)
LOMG (*n*, %)	6 (86%)	7 (64%)	5 (38%)	18 (58%)
Thymectomy (*n*, %)	2 (29%)	4 (36%)	6 (46%)	12 (39%)
Thymoma (*n*, %)	2 (29%)	3 (27%)	2 (15%)	7 (23%)
Other Pathologies (*n*, %)	6 (86%)	7 (64%)	13 (100%)	26 (84%)
IVIG/PLEX Before T0 (*n*, %)	7 (100%)	10 (91%)	11 (85%)	28 (90%)
Myasthenic Crisis Before T0 (*n*, %)	2 (29%)	4 (36%)	2 (15%)	8 (26%)
Pyridostigmine (*n*, %)	3 (42.8%)	9 (81.8%)	13 (100%)	25 (81%)
Steroid-Dose (MG/DAY)	27	13.06	12.65	17.57
NSIST (*n*, %)	4 (57%)	7 (64%)	7 (54%)	18 (58%)
AZA (*n*, %)	3 (42.8%)	5 (45.4%)	5 (38.5%)	13 (42%)
AZA-Dose (mg/day, mean)	116,66	105	110	
Other NSIST (*n*, %)	0 (0%)	2 (18.2%)	2 (15.4%)	4 (13%)
Antibodies				
Seropositive (*n*, %)	7 (100%)	11 (100%)	13 (100%)	31 (100%)
AChR (*n*, %)	7 (100%)	11 (100%)	13 (100%)	31 (100%)
MuSK (*n*, %)	0 (0%)	0 (0%)	0 (0%)	0 (0%)
MGFA Class				
I (*n*, %)	0 (0%)	0 (0%)	0 (0%)	0 (0%)
IIA (*n*, %)	0 (0%)	0 (0%)	2 (15%)	2 (6%)
IIB (*n*, %)	0 (0%)	3 (27%)	5 (38%)	6 (19%)
IIIA (*n*, %)	2 (29%)	1 (9%)	2 (15%)	5 (16%)
IIIB (*n*, %)	4 (57%)	5 (45%)	4 (31%)	13 (42%)
IVA (*n*, %)	0 (0%)	0 (0%)	0 (0%)	0 (0%)
IVB (*n*, %)	1 (14%)	1 (9%)	0 (0%)	2 (6%)
V (*n*, %)	0 (0%)	0 (0%)	0 (0%)	0 (0%)
QMG	18.28	14.2	15.92	16.13
MG-ADL	12.42	10.4	9.69	10.84
MGC	21.6	17.2	16.8	18.53

EOMG: early-onset Myasthenia Gravis; LOMG: late-onset Myasthenia Gravis; IVIg: intravenous immunoglobulin; PLEX: plasma exchange; NSIST: non-steroidal immunosuppressive treatment; AZA: azathioprine; AChR: acetylcholine receptor; MuSK: muscle-specific kinase; MGFA: Myasthenia Gravis Foundation of America; MGC, Myasthenia Gravis Composite; QMG, Myasthenia Gravis quantitative scale.

**Table 2 jcm-14-08205-t002:** Adverse events during treatment with Eculizumab, Ravulizumab, and Zilucoplan.

	Eculizumab	Ravulizumab	Zilucoplan
Adverse Events	3 (43%)	5 (45%)	8 (62%)
Nausea	0 (0%)	3 (27%)	2 (15%)
Fever	0 (0%)	2 (18%)	0 (0%)
Dizziness	0 (0%)	1 (9%)	0 (0%)
General Malaise	0 (0%)	1 (9%)	0 (0%)
Upper Resp Tract Infection	0 (0%)	1 (9%)	0 (0%)
Pneumonia	1 (14%)	1 (9%)	0 (0%)
Bronchitis	0 (0%)	1 (9%)	0 (0%)
Headache	1 (14%)	0 (0%)	4 (31%)
Diarrhea	1 (14%)	0 (0%)	0 (0%)
Skin Rash	0 (0%)	1 (9%)	0 (0%)
Infusion Site Reaction	0 (0%)	0 (0%)	8 (62%)

## Data Availability

The data presented in this study are available on request from the corresponding author due to privacy restrictions.

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
