# Peer review of "Complement Inhibitors in Generalized Myasthenia Gravis: Comparison of Administration Schedules, Efficacy, and Safety"

_jcm, 2025, doi:10.3390/jcm14228205_

Round 1
Reviewer 1 Report
Comments and Suggestions for Authors
Thank you for this interesting manuscript. I have a few suggestions.
- Add the total number of patients and number of each treatment in the abstract. If word count allows, also add the absolute MG-ADL and QMG score values before and after.
- In the introduction section, there is a sentence “Activation of the classical complement pathway plays a crucial role in MG”. Please add AChR MG since other types of MG such as MuSK MG are not affected by complement activation.
- In the introduction section, the sentence “Zilucoplan, furthermore, by binding to C5b, directly prevents interaction to C6 (7–9)” is confusing and should be rephrased.
- Please mention if there were any factors that favored choosing one C5IT over the other (i.e. did patients mentioned preferring Zilucoplan because of the subcutaneous administration, did the treating physician prefer one over the others and why).
- Please specify why MGFA III was needed for eculizumab and not the other two C5IT.
- The paragraph right before statistical analysis should be separated from MGC paragraph. In this same paragraph, you should specify why patients were considered as responders for 3-point and 5-point reductions in MG-ADL and QMG respectively, instead of simply using the MCID cutoffs.
- Over a third of the patients had a thymectomy. Please mention the timing of the thymectomy as it could be a potential confounder for improvement if it was done just before the patients were started on C5IT.
- Which diseases were considered a comorbidity?
- When reporting the results of each of the scales (MG-ADL, QMG etc), include the absolute value of the scale, not just the F ratio and p value. This information could be seen in Figure 2 but it should be explicitly mentioned in the body of the manuscript.
- In the paragraph of steroid-sparing effect, one sentence mentions: “5 patients in the Eculizumab
group were assuming prednisone at baseline”, which appears to be a typographic error.
Reviewer 2 Report
Comments and Suggestions for Authors
The study represents a valuable contribution to clinical neurology, comparing three modern complement inhibitors (eculizumab, ravulizumab, and zilucoplan) in a real-world treatment setting of generalized myasthenia gravis. The research is well-designed, prospective, and longitudinal, with clearly defined inclusion criteria, while the use of MG-ADL, QMG, and MGC scales provides validated and appropriate measures of clinical efficacy. Statistical analysis using a linear mixed model and MCID evaluation is sound and supports reliable interpretation of outcomes. The results demonstrate significant clinical improvement across all treatment groups with no major differences among drugs, accompanied by a notable steroid-sparing effect. The safety profile is favorable, with mostly mild to moderate adverse events consistent with previous clinical trial data. The authors clearly document reductions in MG-ADL, QMG, and MGC scores, confirming the similar mechanism of action and therapeutic benefit of all C5 inhibitors. The main study limitations are the small sample size, population heterogeneity, and variable follow-up durations, which limit generalizability. The discussion is comprehensive but could be strengthened by including pharmacoeconomic considerations and analysis of treatment adherence related to administration schedules. Despite these limitations, the paper is methodologically sound, clinically relevant, and contributes to the growing evidence supporting complement inhibition as an effective and safe therapeutic option for patients with generalized myasthenia gravis.
Reviewer 3 Report
Comments and Suggestions for Authors
This manuscript presents a real-world comparative study of three terminal complement C5 inhibitors (Eculizumab, Ravulizumab, and Zilucoplan) in the treatment of generalized myasthenia gravis (gMG). The study evaluates and compares the administration schedules, efficacy (using MG-ADL, QMG, and MGC scales), and safety profiles of these biologic therapies. The authors report that all three drugs demonstrated comparable and significant clinical efficacy in improving gMG scores and reducing steroid use, with no statistically significant differences observed among them. The safety profiles were also generally similar, with most adverse events being mild to moderate, though some serious events were noted. The conclusion posits these agents as valid options when conventional therapies fail. The manuscript addresses a relevant and timely clinical question regarding the comparison of novel C5 inhibitors in gMG. The real-world design is a valuable addition to the existing literature from randomized controlled trials. The study is generally well-conceived, and the statistical analysis appears appropriate for the primary comparisons.
Major Concerns:
- Sample Size. The relatively small number of enrolled patients remains the major limitation of this study, which may reduce the statistical power and generalizability of the findings. Although the authors acknowledged this issue in the Discussion, further emphasis on its potential impact on data interpretation would be beneficial.
- Novelty and Scientific Contribution. It is not entirely clear what the innovative aspect of this work is. The authors should explicitly highlight how their study differs from previous clinical or observational reports and what new insights it adds to the field of gMG treatment.
- Figure format. Although the figures indicate that there are no significant differences among drugs, they do not clearly illustrate within-group changes across time points. This makes it difficult for readers to visualize longitudinal effects without referring back to the text. Please consider improving figure clarity and including relevant statistical annotations.
- Discussion Section. The discussion lacks clear structure and contains lengthy paragraphs that make it difficult to follow. It would benefit from clearer organization, with distinct subsections. Conciseness and thematic focus would improve readability.
- Other concerns:
The letter P in statistical results should be presented in uppercase and italicized (i.e., P < 0.05) throughout the manuscript.
Tables should be uniformly formatted in a standard three-line table style for consistency and clarity.
The error bars use “+/-”; the standard “±” symbol would be preferable.
